# Evaluating Different Factors That Affect the Nesting Patterns of European and Algerian Hedgehogs in Urban and Suburban Environments

**DOI:** 10.3390/ani13243775

**Published:** 2023-12-07

**Authors:** Héctor Gago, Robby M. Drechsler, Juan S. Monrós

**Affiliations:** Instituto Cavanilles de Biodiversidad y Biología Evolutiva, Universitat de València, C/Catedràtic José Beltrán Martínez, 2, 46980 Paterna, Spain; robby.drechsler@uv.es (R.M.D.); juan.monros@uv.es (J.S.M.)

**Keywords:** habitat use, mammals, Spain, spatial ecology, Valencia

## Abstract

**Simple Summary:**

Hedgehogs are commonly found inhabiting urban environments; however, it is still unclear how human presence and activity affects their biology. In this study, we focus on their nesting behavior in the eastern Iberian Peninsula. We radio-tracked 30 male hedgehogs in the vicinity of Valencia and recorded where and how they build their nests. We found that hedgehogs seem to avoid areas with a high human presence, as their nests were mainly concentrated in small green patches with bush-like vegetation (like hedges). We used two different hedgehog species (the European and the Algerian Hedgehog), but we could not evidence differences between the species or other groupings we considered. We even recorded cases where the same nests were being used by two males of both species. Our results indicate that a correct management of forest patches within and near human settlements, for example, by not removing bush vegetation and increasing the connectivity between those patches by ecological corridors, could significantly improve hedgehog conservation.

**Abstract:**

Small undisturbed patches in urban environments serve as important refuges for wildlife, e.g., hedgehogs. However, the effects of urbanization on certain biological aspects, like nesting behavior, remain unknown. We captured and tracked the movement of 30 male hedgehogs of two co-existing species: Algerian and European hedgehogs. The study was carried out in Valencia (eastern Spain). We distinguished six macrohabitats and five subhabitats. We analyzed the proportions of the macro and subhabitats where nests were found to calculate a resource selection function and fit GLMs. Hedgehog nests tended to concentrate in areas with little human disturbance and were built in hedges or under bush-like vegetation. We did not find any significant differences between species or other considered groups. We noted that nests were distributed around hard-to-find suitable habitat patches. We even recorded a case of two males from both species simultaneously using one nest. Our results suggest that hedgehog conservation in urban environments can be improved by the correct management of forest patches by conserving bush-like vegetation and improving the connectivity between suitable patches with ecological corridors.

## 1. Introduction

Urban development and the consequent fragmentation of surrounding habitats are a global problem for species that live close to cities [1]. Land-use transformation and changes in vegetation structure are responsible for the eradication of many native plants and animals in urban environments [2]. One of the major changes that species are subject to is reduced dispersion and isolation, which lead to a low gene flux that can favor endogamy [3]. Thus, providing a wide range of usable ecotones for species is vitally important for their conservation. In urban environments, green areas are extremely important for maintaining urban biodiversity and human welfare [4]. Such areas are refuges where many species can escape some of the biotic and abiotic pressures present in other habitats and/or may offer them new opportunities [5]. Many studies have revealed the impact of urbanization on micromammal species [6,7,8]. However, only a few studies have focused on hedgehogs, like the European Hedgehog (*Erinaceus europaeus* Linnaeus, 1758) or the Algerian Hedgehog (*Atelerix algirus* Lereboullet, 1842). Finally, no studies have analyzed the effect of urbanization on nesting patterns in areas where both the aforementioned hedgehog species co-exist [9,10].

The European Hedgehog is a small insectivore (~30 cm long and weighing 800 g) that is distributed throughout the Iberian Peninsula and in parts of central Europe, and the northern edge of its distribution reaches Finland and Norway [11]. This mammal has adapted to numerous different habitats ranging from forests to mountainous, and rural and even urban environments. The Algerian Hedgehog is slightly smaller (~21 cm long and weighing 550 g) and is lighter in color than its congener. Its distribution goes from North Africa and passes through the Iberian Peninsula (only on the eastern coast) to south France. It is also present on the Canary and Balearic Islands, where two subspecies are described, namely *A. algirus caniculus* and *A. algirus vagans*, respectively.

Hardly any studies describe hedgehog nests and nesting patterns, and the few there are have been carried out mainly in the UK [12,13,14,15], Ireland [16] or Finland [17], with no studies in the Iberian Peninsula for either the European or the Algerian Hedgehog. A recent study in Spain [10] indicates how different factors affect the habitat use and spatial ecology of hedgehogs. Analyzing the nesting patterns of these individuals would considerably supplement those results, especially considering that this would be the first study on hedgehog nesting patterns in this region.

Some studies demonstrate that European Hedgehogs nest preferably close to buildings and villages, and have a wider variety of secure nesting habitats compared to croplands or forest patches [17]. Other studies reveal that hedgehogs tend to nest close to bushes and pathways, but avoid pastures [16,18,19], while they preferably use forests for hibernation [18,20]. Within its movement area, a hedgehog has several diurnal nests and the intensity at which it visits them varies [13]. Diurnal nests can also be abandoned and later reused for hibernation [13,16,21]. The habitat preference of the European and Algerian Hedgehogs varies from region to region, but most studies indicate that forests are the most frequently used vegetal structures by European Hedgehogs [10,22,23]. Golf courses also seem to be a favorable habitat where hedgehog population density can be relatively high [24]. There is only one study about the Algerian Hedgehog, which indicates that the optimal biotope for this species is characterized by open spaces and abundant herbaceous cover [25].

In Valencia (eastern Spain), there are areas in which both hedgehog species co-exist, i.e., peri-urban areas. A previously published study by the authors [10] indicates that the hedgehogs in the study area preferably use pine forest patches, low-density urban areas (like abandoned buildings) and habitats with abundant bush-like vegetation. The results in this study also reveal that larger hedgehogs tend to have wider home ranges, which could also mean that they use more nests.

The aim of this study was to analyze which factors influence the nest distribution and use of European and Algerian Hedgehogs in areas with a high degree of anthropization. We hypothesized that there would be no differences between species and that the habitat fragmentation induced by urban development would lead to a denser nest distribution or more intense nest use (i.e., the same nest employed by more hedgehogs). We also expected hedgehog nests to tend to concentrate near more naturalized habitats (like forest patches) and that areas with marked human presence would be avoided.

## 2. Material and Methods

### 2.1. Study Area

The study was carried out in the Spanish Autonomous Valencian Community (eastern Iberian Peninsula) (Figure 1). The city of Valencia has about 800,000 habitants, a mean temperature of 17.4 °C and annual accumulated precipitations of 445 mm, which mainly fall between October and April [26]. The study area (Figure 1) is located in a peri-urban area in the municipality called Godella (0°25′14.9″ O; 39°31′34.5″ N), and covers 60 ha. This area presents patches of Aleppo Pine (*Pinus halepensis*), high- and low-density urbanizations, growing orange crops, abandoned croplands and a ravine.

### 2.2. Fieldwork

Thirty male hedgehogs were radio-tracked between March 2017 and May 2019. Females were not included in this study because it is known that they tend to significantly reduce their home ranges and movements during the reproductive period, and given the relatively low sample size, could induce an important bias in data [27]. We captured 18 hedgehogs (10 European, 8 Algerian) and released 12 rehabilitated hedgehogs (9 European, 3 Algerian). The latter had been taken to the “La Granja” wildlife center in El Saler due to multiple causes. All the individuals were originally from the metropolitan area of Valencia. The individuals were kept in captivity for between 1 and 2 weeks.

The capturing of individuals was conducted following the same methodology described in [10]. Briefly, we carried out nocturnal transects with bright flash lights (Nitecore, Guangzhou, China, 1000 lumens) and used 12 Havahart rabbit traps (81 × 25 × 30 cm) baited with peanut butter [19,28]. All the captured or rehabilitated individuals were sexed, weighed and marked when stress induced by capturing and manipulation allowed this, and they were all in good condition (not rolled up and/or hissing) following the criteria of Bunnell [29].

The tracking of individuals was carried out by radio-tracking. We attached a VHF radio emitter (Pip3 Biotrack, Wareham, UK) to 30 hedgehogs by gently cutting spines and gluing the device with a putty-like product by Ceys (Barcelona, Spain). The individuals were tracked once per day for 1 month. The individuals were approached by the homing technique [30] after visually detecting them. We recorded the UTM coordinates using a GPS device (Garmin etrex 20, Barcelona, Spain). For more detailed information on this methodology, see [10]. None of the attached devices weighed more than 4% of the hedgehog’s body weight [31] and they were all gently removed after the monitoring time.

### 2.3. Habitat Characterization

Using maps and the visual characterization of the study site, we distinguished six macrohabitat types and five subhabitat types. We took the macrohabitat to be the general environment that nests were built in, especially when looking at the vegetation structure of the surrounding areas. The discerned macrohabitats were the following: High-Density Urban (HDU), characterized by actively used human settlements; Low-Density Urban (LDU), areas like abandoned buildings with a higher degree of bush vegetation; Abandoned Crops (AC), former croplands with dense bush vegetation; Growing Crops (CU), crops being grown, mainly orange plantations, with regular human presence; Pine Woods (PW), patches of forests with arboreal vegetation dominated by Aleppo Pine and scrubs; and Ravine (RV), an abruptly lowering terrain level with dense bush vegetation. As a subhabitat, we considered the materials that nests were built with/in. The distinguished subhabitats were the following: Hedges (HED), including bushes, scrubs and brambles; Leaves (LEA), accumulations of leaf litter from different trees and bushes; Roots (ROO), small cavities between tree roots; Rubbish (RUB), cavities in accumulations of rocks, bricks and other materials left by humans; and Walls (WALL), cavities and holes in walls of human constructions.

### 2.4. Data Analysis

In order to analyze whether the observed distribution of macrohabitat types significantly differed from randomness, we calculated a resource selection function (RSF) following the work of [32] by randomly generating a number of points within each Kernel 90 home range (see [10] about how home ranges were calculated), which equaled an individual’s number of real localizations, and by assigning the macrohabitat each point (random and real) was located in. These random points were considered a measure of habitat availability. Then we fitted a binomial GLM by comparing the habitat proportions of the real and random points. As subhabitats are not drawn on a map, we applied a slightly modified methodology. We assigned a number (1–5) to each subhabitat and let R pick random numbers between 1 and 5 for a number of times that equaled the real subhabitat recordings for each individual. Hence, we had a random and real dataset to compare by applying the same methodology used for macrohabitats.

To analyze how nest distribution varied in each macrohabitat and subhabitat type, we considered the following explanatory variables: species (European and Algerian); origin (native or translocated); weight upon release (individuals grouped in two categories: <700 g and >700 g); and time of year (considering two periods: cold [November–March] and warm [April–October]).

For each individual, we estimated the proportion of nests located in each macro and subhabitat. Given that percentages do not have normal distribution, for the statistical analysis, they were transformed using arcsine transformation [33] after applying the following formula, where *p*′ was the transformed value and *p* was the original value (expressed as a proportion from 0 to 1):$$p’=\frac{arcsin\left(\sqrt{p}\right)*180}{\pi }$$

Next, we fitted a GLM with normal error distribution and an identity linking function, and took the transformed habitat proportions as the independent variable and the following predictors: species, origin, weight and period. We also tested the parametric assumptions of residual normality and homoscedasticity by the Shapiro–Wilk test and the Breush–Pagan test, respectively. To analyze the differences in habitat proportions in each group (species, origin, weight and period), we fitted a GLM with normal error distribution and an identity linking function, and carried out a post hoc Tukey test to see which habitats were used significantly differently [34].

Mapping and habitat characterization were completed with the QGIS V2.18 software [35]. All the statistical analyses were carried out using the R v4.2.2 software [36]. The employed packages were “lme4” for fitting GLMs, “lmtest” for testing parametric assumptions and “multcomp” for Tukey tests.

## 3. Results

We detected 230 hedgehog nests. In four cases, we observed that two hedgehogs simultaneously occupied one nest. For these two cases, both species were found in the same nest. The nest distribution between species and origins is summarized in Table 1.

### 3.1. Macrohabitats

The RSF for macrohabitats showed that their distribution significantly differed from randomness (GLM, Z value = 7.16, *p* < 0.01). The analysis of the differences in the macrohabitat proportions between the distinct groups (species, origins, weight and period) did not show any significant differences in most cases. Exceptionally, we found significant differences in the LDU habitat type, which was used by a higher proportion of translocated individuals than by native ones, and also by a higher proportion during the warm period than the cold one (Table 2).

The Tukey test showed that hedgehogs clearly used PW significantly more than the other habitats in all the groups (Table 2 and Table 3). Furthermore, during the cold period, for the hedgehogs weighing <700 g and native hedgehogs, a significant preference for LDU environments was observed over CU (Table 2 and Table 3). In native hedgehogs, LDU was also used significantly more than the RV and HDU habitats (Table 2 and Table 3).

### 3.2. Subhabitats

The resource selection function in the subhabitats showed that their distribution also significantly differed from randomness (GLM, *Z* value = 7.349 *p* < 0.01). The analysis of the differences in the subhabitat proportions among groups showed only significant differences between species, where European Hedgehog nests were found mostly in HED, but the Algerian Hedgehog nests were more distributed in the other subhabitat types (Table 4).

The Tukey test revealed that hedges were used significantly more than the other subhabitats for all the groups (Table 3 and Table 4). Additionally, hedgehogs weighing <700 g and the Algerian Hedgehogs significantly preferred LEA over human structures (WALL) (Table 3 and Table 4).

## 4. Discussion

### 4.1. General

Our results find that more than 50% of nests were located in PW and HED. Hedgehogs select a wide variety of habitats for nesting [14] or prefer to nest near rural human settlements on croplands [37]. Nesting behavior even changes seasonally. In places like Finland, hedgehogs prefer to nest near residential areas during mating and postmating periods. During prehibernation and hibernation, nests are preferably built in forest areas [17]. Our results indicate that in anthropized areas, where more limiting factors affect hedgehog home ranges (i.e., roads, buildings, fences, noise, etc.), they tend to use a wider variety of habitats to nest depending on their ability to travel between suitable patches in fragmented environments. Thus, the variety of habitats is an essential element for providing feeding grounds, protection against predators and suitable areas for reproduction [17,19,20,37,38].

### 4.2. Macrohabitats

Urban environments represent a large portion of the study area. It is known that hedgehogs enter human settlements and gardens, which they are able to use them as refuges [39]. However, they are not the preferred habitat type in our study. In our case, wastelands, LDU environments and PW patches are the preferable habitats for nesting in HDU environments. This can be explained by a better trade-off between resource and refuge density and human disturbance in these areas. This coincides with the results reported in other studies, which state that hedgehogs tend to avoid human disturbance [40]. In our study area, CU and RV represent a relatively small portion of the available area. However, these areas are used as connections between preferred habitats [10].

The differences observed between translocated and native individuals can be explained by a “releasing effect” [10], whereby the hedgehogs introduced into a new environment tend to make longer exploratory movements [41] and subsequently use more nests. In fact, the results from other studies show that translocated hedgehogs can travel up to 4 km on 19 days [23] or up to 5 km on 108 days [42]. Furthermore, as these animals were left in captivity during their rehabilitation in the wildlife center, they could have gotten used to human presence and, hence, could be more inclined to seek areas with marked human presence than native hedgehogs would. Indeed, this tendency can be observed from our results, although differences were not significant.

We also found differences between the two considered times of the year. The LDU areas presented significantly more hedgehog nests during the warm period (Apr–Oct). This period includes the maximum activity and the mating season [11]. These habitats are employed to establish intermediate resting points when traveling between foraging and reproduction areas. Thus, we encourage future research to focus on assessing the conservation potential and to value these habitat types.

### 4.3. Subhabitats

Our results showed a clear preference for building nests in HED over the other subhabitat types. HED and bushes play a fundamental role in maintaining biodiversity [43] by providing refuges and being used as ecological corridors by many species like birds [44] or invertebrates [45]. The authors of one study indicate that HED and bushes with good floor cover favor biodiversity by providing suitable nesting sites for hedgehogs, as well as LEA where potential prey like snails and coleopterans can be found [46]. Indeed, in some cases involving hedgehogs weighing <700 g and the Algerian Hedgehogs, we also detected a significant preference for building nests in LEA compared to human structures where human disturbances can often be expected.

Interestingly, we noted significant differences between the Algerian and European Hedgehogs in their use of all the subhabitats, except for human structures. The Algerian Hedgehog preferably used LEA, ROO and RUB, while the European Hedgehogs employed more HED. This is in accordance with the results reported in other studies, where European Hedgehogs clearly prefer nests in HED and bushes [12,16,24]. There are only a few studies about the Algerian Hedgehog, which show its preference for nesting in cactuses or tree holes [47]. This in fact contrasts with the results that we obtained in a previous study [10], which found that European Hedgehogs have a more generalist character regarding habitat use than Algerian Hedgehogs. However, other studies reveal that Algerian Hedgehogs in the Iberian Peninsula also use other types of environments like grasslands and scrublands [25]. This allows us to think that Algerian Hedgehogs may not be a very strict habitat specialist and may have a certain tolerance level, at least when considering different aspects of habitat use. This also shows that more research needs to be done with individuals of both sexes, and also with juveniles, to further understand the different facets of habitat use.

### 4.4. Nest Sharing and Parasites

The high habitat fragmentation level in the urban environment leads to a high concentration of nests in scarcely distributed suitable habitat patches. It is known that hedgehogs clearly tend to avoid roads [48]. Another recent study indicates that hedgehogs in highly fragmented urban environments show subtle changes in behavior compared to hedgehogs from less fragmented areas [49]. In the Netherlands, Bergers and Nieuwenhuizen [50] studied the viability of hedgehog populations. They indicate that the first limiting factor is the size and quantity of suitable habitat patches, followed by the presence of roads.

It is known, and also herein observed, that hedgehogs eventually share nests simultaneously [13,16,19] or not (another hedgehog enters once the previous one has left) [12], although they are generally solitary animals. This, combined with the spatial limitation of every habitat patch in the urban matrix, could favor parasite transmission between hedgehogs. Actually, there are many pathogens to be found in/on hedgehogs: fungi like *Cladosporium* or *Rhizopus* [51]; bacteria like *Salmonella* [52], *Staphylococcus lentus* [53], *Escherichia coli* [54], *Rickettsia felis* [55] or *Leptospira interrogans* [56]; and viruses like rabies or herpesvirus [57]. We also recently showed that hedgehogs can act as potential vectors for important zoonotic diseases, which may endanger the welfare of humans and pets [10,58,59]. Thus, we encourage further research that focuses on the role of nest distribution and nest sharing in the transmission of parasites and diseases in hedgehog populations.

## 5. Conclusions

Our results support the proposed hypothesis. Both European and Algerian Hedgehogs tend to preferably use areas with little human disturbance and require forest patches to nest, mostly bush-like formations and LEA for nest building. Urban environments are highly fragmented habitats that can be used by hedgehogs, but also pose serious threats for them. Basic ecological information to improve hedgehog conservation is urgently required because hedgehog populations are rapidly declining in Europe [28]. Our results suggest that hedgehog conservation in urban environments can be improved in two ways: on the one hand, by maintaining hedgehog populations in peri-urban environments, improving the management of forest patches and conserving bush-like vegetation; and on the other hand, by improving the connectivity between suitable patches by installing ecological corridors.

## Figures and Tables

**Figure 1 animals-13-03775-f001:**
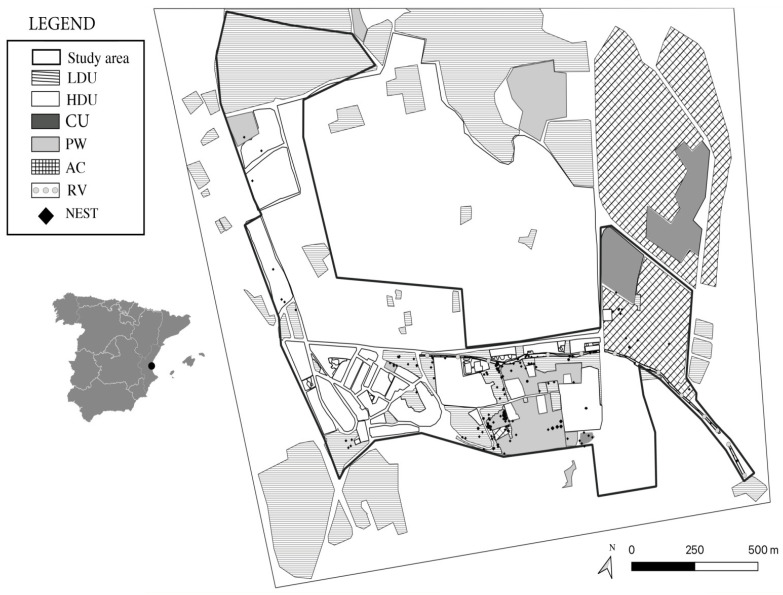
Location of the study area in the Iberian Peninsula. Macrohabitats are distinguished as follows: High-Density Urban (HDU); Low-Density Urban (LDU); Abandoned Crops (AC); Growing Crops (CU); Pine Woods (PW); and Ravine (RV). Hedgehog nests are represented as black diamonds.

**Table 1 animals-13-03775-t001:** Number of individuals (N), mean ± SE of the number of nests per individual (L) and the range of values (Range) and mean ± SE of hedgehog weight. Data are grouped by species, origins and study sites.

	Species		Origin	
	European Hedgehog	Algerian Hedgehog	Indigenous	Translocated
N	14	17	18	13
L	6.92 ± 0.48	7.21 ± 0.66	6.67 ± 0.27	7.2 ± 0.39
Range	6–8	6–8	6–8	6–8
Weight (g)	769.82 ± 223.34	545.07 ± 82.51	643.61 ± 155.71	702.54 ± 263.95

**Table 2 animals-13-03775-t002:** Mean ± SD of the percentages of nests in each macrohabitat type for the different groups. The GLM results: degrees of freedom and sample size (*Df*|n), Z statistic and significance level (*P*). Significant results are highlighted in bold. Abbreviations: High-Density Urban (HDU); Low-Density Urban (LDU); Abandoned Crops (AC); Growing Crops (CU); Pine Woods (PW); and Ravine (RV).

	Indigenous	Translocated	*Df|n*	*Z*	*P*	European	Algerian	*Df|n*	*Z*	*P*
PW	57.9 ± 32.5	51.5 ± 42.5	1|28	0.97	0.34	54.6 ± 38.4	56.3 ± 24.7	1|28	0.70	0.48
HDU	4.6 ± 13.3	11.8 ± 26.4	1|28	0.67	0.5	7.6 ± 22.7	7.3 ± 15.4	1|28	−1.34	0.19
**LDU**	**25.8 ± 27.8**	**11.1 ± 17.9**	**1|28**	**−2.89**	**<0.01**	18.8 ± 23.7	21.5 ± 27.6	1|28	0.69	0.49
CU	1.2 ± 5.2	4.8 ± 16.5	1|28	0.34	0.73	3.4 ± 13.9	1.7 ± 6.2	1|28	−0.16	0.86
AC	6.4 ± 10.7	14.9 ± 20.4	1|28	1.11	0.27	8.7 ± 12.6	11.3 ± 19.3	1|28	−1.58	0.12
RV	3.4 ± 9.2	6.0 ± 14.2	1|28	0.13	0.89	7.1 ± 14.3	1.0 ± 3.5	1|28	1.22	0.23
	**<700 g**	**>700 g**	** *Df|n* **	** *Z* **	** *P* **	**Warm**	**Cold**	** *Df|n* **	** *Z* **	** *P* **
PW	63.9 ± 35.9	40.5 ± 33.3	1|28	1.75	0.09	54.7 ± 32.2	56.1 ± 42.4	1|28	1.40	0.17
HDU	3.0 ± 28.4	15.1 ± 28.4	1|28	−1.66	0.1	10.5 ± 23.5	3.4 ± 13.3	1|28	0.08	0.93
LDU	20.2 ± 25.0	19.5 ± 25.0	1|28	0.01	0.98	**15.5 ± 15.9**	**25.7 ± 33.4**	**1|28**	**−2.37**	**0.02**
CU	1.2 ± 17.2	5.2 ± 17.2	1|28	−0.27	0.78	3.4 ± 13.9	1.7 ± 6.2	1|28	0.10	0.91
AC	8.9 ± 15.8	11.4 ± 15.8	1|28	−1.83	0.07	12.6 ± 19.2	6.1 ± 8.4	1|28	0.38	0.70
RV	2.2 ± 16.0	8.3 ± 16.0	1|28	0.3	0.76	3.3 ± 10.6	6.0 ± 12.4	1|28	0.18	0.85

**Table 3 animals-13-03775-t003:** Results of the post hoc Tukey tests showing the significance (*P*) of the differences in the percentages of nests between the different macro and subhabitats for each group (only significant results (*p* < 0.05) are shown). Abbreviations: Hedges (HED); Walls (WALL); Rubbish (RUB); Leaves (LEA); Roots (ROO); High-Density Urban (HDU); Low-Density Urban (LDU); Abandoned Crops (AC); Growing Crops (CU); Pine Woods (PW); and Ravine (RV).

Group	Macrohabitat	*P*	Subhabitat	*P*
Warm	PW–AC	<0.001	HED–WALL	<0.001
	PW–CU	<0.001	HED–RUB	<0.001
	PW–RV	<0.001	HED–LEA	<0.001
	PW- LDU	<0.001	HED–ROO	<0.001
Cold	PW–AC	<0.001	HED–WALL	<0.001
	PW–CU	<0.001	HED–RUB	<0.001
	PW–LDU	0.001	HED–LEA	<0.001
	PW–RV	<0.001	HED–ROO	<0.001
	PW–HDU	<0.001	-	-
	LDU–CU	0.001	-	-
>700 g	PW–AC	<0.001	HED–WALL	<0.001
	PW–CU	<0.001	HED–RUB	<0.001
	PW–RV	<0.001	HED–LEA	<0.001
	PW–HDU	<0.001	HED–ROO	<0.001
	PW–LDU	0.006	-	-
<700 g	PW–AC	<0.001	LEA–WALL	0.049
	PW–CU	<0.001	HED–WALL	<0.001
	PW–LDU	<0.001	HED–RUB	<0.001
	PW–RV	<0.001	HED–LEA	0.038
	PW–HDU	<0.001	HED–ROO	<0.001
	LDU–CU	<0.001	-	
Native	PW–AC	<0.001	HED–WALL	<0.001
	PW–CU	<0.001	HED–RUB	<0.001
	PW–RV	<0.001	HED–LEA	<0.001
	PW–HDU	<0.001	HED–ROO	<0.001
	PW–LDU	<0.001	-	-
	LDU–AC	0.021	-	-
	LDU–CU	<0.001	-	-
	LDU–RV	0.001	-	-
	LDU -HDU	0.001	-	-
Released	PW–AC	0.007	HED–WALL	<0.001
	PW–CU	<0.001	HED–RUB	<0.001
	PW–RV	<0.001	HED–LEA	<0.001
	PW–HDU	<0.001	HED–ROO	<0.001
	PW–LDU	0.001	-	-
European	PW–AC	<0.001	HED–WALL	<0.001
	PW–CU	<0.001	HED–RUB	<0.001
	RV–PW	<0.001	HED–LEA	<0.001
	PW–HDU	<0.001	HED–ROO	<0.001
	PW–LDU	<0.001	-	-
Algerian	PW–AC	<0.001	HED–WALL	<0.001
	PW–CU	<0.001	HED–RUB	<0.001
	RV–PW	<0.001	HED–ROO	<0.001
	PW–HDU	<0.001	LEA–WALL	0.035
	PW–LDU	<0.001		

**Table 4 animals-13-03775-t004:** Mean ± SD of the percentages of nests in each subhabitat type for the different groups. The GLM results: degrees of freedom and sample size (*Df*|n), *Z* statistic and significance level (*P*). Significant results are highlighted in bold.

	Indigenous	Translocated	*Df|n*	*Z*	*P*	European	Algerian	*Df|n*	*Z*	*P*
HED	52.1 ± 23.4	58.9 ± 22.8	1|28	0.11	0.91	**62.9 ± 22.9**	**44.2 ± 19.0**	**1|28**	**6.42**	**<0.01**
LEA	20.7 ± 19.6	17.8 ± 19.8	1|28	−0.10	0.91	**14.1 ± 18.7**	**26.7 ± 18.5**	**1|28**	**−7.43**	**<0.01**
ROO	9.3 ± 12.2	5.8 ± 14.9	1|28	−1.47	0.15	**5.3 ± 13.2**	**11.2 ± 13.0**	**1|28**	**−4.81**	**<0.01**
RUB	9.7 ± 18.7	5.2 ± 12.1	1|28	0.74	0.46	**3.7 ± 10.4**	**13.4 ± 21.0**	**1|28**	**−7.97**	**<0.01**
WALL	7.1 ± 13.9	9.1 ± 19.7	1|28	0.46	0.64	8.4 ± 18.0	7.2 ± 14.2	1|28	−5.23	0.54
	**<700 g**	**>700 g**	** *Df|n* **	** *Z* **	** *P* **	**Warm**	**Cold**	** *Df|n* **	** *Z* **	* **P** *
HED	49.8 ± 22.6	63.3 ± 22.1	1|28	2.02	0.09	56.2 ± 23.9	53.0 ± 22.6	1|28	0.34	0.73
LEA	24.8 ± 19.5	10.5 ± 16.1	1|28	0.83	0.41	20.2 ± 19.4	18.7 ± 20.2	1|28	−0.04	0.96
ROO	8.7 ± 14.4	6.5 ± 11.4	1|28	−0.24	0.80	4.2 ± 9.5	12.7 ± 16.0	1|28	−1.46	0.16
RUB	9.8 ± 18.9	4.6 ± 10.5	1|28	0.70	0.48	8.8 ± 18.9	6.6 ± 12.7	1|28	0.99	0.33
WALL	4.9 ± 12.1	12.9 ± 21.3	1|28	−0.88	0.38	8.6 ± 18.4	6.9 ± 13.4	1|28	0.25	0.80

## Data Availability

Data are not publicly available, anyone interested in using the data of this study, please contact the corresponding author of the article.

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
