# Peer review of "Evaluating Different Factors That Affect the Nesting Patterns of European and Algerian Hedgehogs in Urban and Suburban Environments"

_animals, 2023, doi:10.3390/ani13243775_

Round 1

Reviewer 1 Report

Comments and Suggestions for Authors

There is some interesting data here giving an insight into the habitats used by hedgehogs for nesting. However, I do not see there is sufficient for the question proposed and for the focus of the study. 

I have some concerns regarding the sample size in this study which affects the scientific soundness - the study is very heavily skewed towards European hedgehogs yet the main focus of the study is a comparison of the two species. There are only eight established Algerian hedgehogs studied, as it cannot be known to what extent the three released animals are behaving naturally, and their range/nesting behaviour is likely to fluctuate considerably when newly released. It is not clear how long they were released for until data started being collected, nor how long they were in captivity, site of origin, and for what condition (which will all affect their behavior upon release). I struggle with the inclusion of the rehabilitated hedgehogs for these reasons, which bring the actual comparison down to 10 European and eight Algerian hedgehogs. The remaining European hedgehogs were at a different site. It is unclear how much all individuals roamed across the site, and how many different habitats each individual would have access to. 

As the golf course wasn't included in the main analysis of the study it is unclear why these data have been presented. I think it overly confuses the study. This is of particular challenge as only 4 days of data were collected in this habitat, compared to one month in the periurban habitat. Hedghog nest movement can fluctuate with season, weather, habitat etc. and considerably by individual. 

As the data have been amalgamated for all translocated hedgehogs, yet there are differences between the origins identified, this analysis lacks integrity. Likewise, as differences are identified between species, to then amalgamate these invalidates the analysis. 

I would also question the sensitivity of the GPS tags - is this sufficent to determine the habitat type that the golf course hedgehogs were nesting in? As accuracy is typically >3 metres in my experience this is unlikely to be adequate in mixed habitats. The variation in locating nests between the two sites is problematic as a result of this. 

For these reasons, a simple comparison between the 10 European hedgehogs and 8 Albanian hedgehogs would have greater robustness and present a clearer study, however this does limit data analysis to some extent, and should therefore be kept simple, rather than the use of GLM with so many variables included. 

As noted by the authors, there is limited published data regarding the nesting behaviour of European hedgehogs in the southern part of their range, so this data set has merit for this focus, and offers value. The short -term nature of much of this (4 days in the golf course) does present some challenges though, so if this could be expanded this value would increase substantially. This is particularly relevant during different seasons, as much of the research published is in much cooler parts of the range. 

Some further comments to support the development of your paper are as follows:

Including more specific detail re: size of hedgehogs would be beneficial in the introduction - rather than just describing as small mammals and a bit smaller. 

It would be helpful to see a map of the nest locations across the two sites, to see how they are distributed. 

Figure 1: The key needs definition - it is not clear what the codes mean, e.g. LDU, HDU. They should also be standardised for formatting, such as the font used and the shading type used for each - e.g. PW.  These are currently on the following page (L134-138) so not obvious to the reader. 

Section 2.2 - The breakdown of hedgehogs equals 73, although 72 is stated on the first line. As 12 were rehabilitated it isn't correct to say on the first line that 72(73) were captured - perhaps studied/observed/radiotracked? 

Spelling/wording errors - e.g. L112 & L242 - weighted/weighting; L145 Rubish; L163 at the moment of the year - should read the time of year or season. L164 - Apr not Abr; L185 & L259 - built not build. 

L118 & L123 - The attachment of GPS tags is described but not VHF tags

L128 - how can you ensure optimal health and stress conditions? How was this measured? I would suggest this can only be determined by veterinary examination. Perhaps this should be reworded if this isn't what was done.

L207 - RSF needs to be defined for clarity.  

L218-223 - the use of abbreviations makes this rather confusing. I found myself needing to refer back pages to know what was being discussed.

L222/L240 - *used significantly more

Comments on the Quality of English Language

Whilst much of the paper is clearly written there are some areas where this could be improved for consistency and for clarity. Examples are provided above. 

Author Response

There is some interesting data here giving an insight into the habitats used by hedgehogs for nesting. However, I do not see there is sufficient for the question proposed and for the focus of the study.

Thank you for your review and comments, find our answers to each one in bold.

I have some concerns regarding the sample size in this study which affects the scientific soundness - the study is very heavily skewed towards European hedgehogs yet the main focus of the study is a comparison of the two species. There are only eight established Algerian hedgehogs studied, as it cannot be known to what extent the three released animals are behaving naturally, and their range/nesting behaviour is likely to fluctuate considerably when newly released. It is not clear how long they were released for until data started being collected, nor how long they were in captivity, site of origin, and for what condition (which will all affect their behavior upon release). I struggle with the inclusion of the rehabilitated hedgehogs for these reasons, which bring the actual comparison down to 10 European and eight Algerian hedgehogs. The remaining European hedgehogs were at a different site. It is unclear how much all individuals roamed across the site, and how many different habitats each individual would have access to.

We respectfully disagree with the statement that the study is “very heavily” skewed towards European Hedgehogs. The main analyses were made only on the urban population which in our case included 19 European and 11 Algerian Hedgehogs. This is also due to differences in abundance of both species, being European Hedgehogs more abundant in this area (pers. obs.). Further, the focus of the study is not the comparison of both species (this is just one of the variables analysed), but to better understand how hedgehogs nest in urban environments and which factors could be affecting it in our latitudes.

Regarding the inclusion of rehabilitated individuals, we already studied the spatial behaviour of the same individuals used in the present study (see the citation below). Indeed, we found that translocated individuals tended to move more and have larger home ranges. This is why we decided to include them here, because this work is a good complement to our previously published article, analysing something more specific than just general spatial ecology. In fact, it is interesting that here, considering nest distribution, we did NOT find differences between translocated and indigenous hedgehogs (except for low-density urban macrohabitat). If rehabilitated individuals would use nests very different than established ones, we would have seen more differences (and you could argue that those differences distort results from other groupings). In conclusion, removing rehabilitated individuals would remove a lot of value from the article, and considering that this is literally the first study on hedgehog nesting behaviour in the Iberian Peninsula, we do not think that this is justified.

Regarding your punctual concerns: data collection began just after release (on a daily routine), the rehabilitated individuals were kept one to two weeks in captivity and released in good health conditions, all individuals were from the metropolitan area of Valencia. We included this information for clarity.

Gago, H., Drechsler, R. M., & Monrós, J. S. (2023). Algerian and European hedgehogs cohabiting in periurban environments: spatial behaviour and habitat use. European Journal of Wildlife Research, 69(1), 19.

As the golf course wasn't included in the main analysis of the study it is unclear why these data have been presented. I think it overly confuses the study. This is of particular challenge as only 4 days of data were collected in this habitat, compared to one month in the periurban habitat. Hedghog nest movement can fluctuate with season, weather, habitat etc. and considerably by individual. As the data have been amalgamated for all translocated hedgehogs, yet there are differences between the origins identified, this analysis lacks integrity. Likewise, as differences are identified between species, to then amalgamate these invalidates the analysis. I would also question the sensitivity of the GPS tags - is this sufficent to determine the habitat type that the golf course hedgehogs were nesting in? As accuracy is typically >3 metres in my experience this is unlikely to be adequate in mixed habitats. The variation in locating nests between the two sites is problematic as a result of this.

After careful consideration and discussion, we agree that including the data from the golf course adds too much confusion in relation to the value of the analysis. Thus we decided to remove everything related to the golf course data and the comparison of nest distances between study sites.

For these reasons, a simple comparison between the 10 European hedgehogs and 8 Albanian hedgehogs would have greater robustness and present a clearer study, however this does limit data analysis to some extent, and should therefore be kept simple, rather than the use of GLM with so many variables included.

We think you meant Algerian Hedgehogs. As mentioned previously we disagree on removing rehabilitated individuals from the analysis.

As noted by the authors, there is limited published data regarding the nesting behaviour of European hedgehogs in the southern part of their range, so this data set has merit for this focus, and offers value. The short -term nature of much of this (4 days in the golf course) does present some challenges though, so if this could be expanded this value would increase substantially. This is particularly relevant during different seasons, as much of the research published is in much cooler parts of the range.

Thank you for the comments. Indeed, data is limited, which is another argument to not remove rehabilitated hedgehogs from our analyses. The data from the golf course have been removed. Our dataset includes an analysis considering the phenology, at least comparing cold and warm periods, which would be interesting to compare with results from other locations with different climatic conditions.

Some further comments to support the development of your paper are as follows:

Including more specific detail re: size of hedgehogs would be beneficial in the introduction - rather than just describing as small mammals and a bit smaller.

Added

It would be helpful to see a map of the nest locations across the two sites, to see how they are distributed.

Figure 1: The key needs definition - it is not clear what the codes mean, e.g. LDU, HDU. They should also be standardised for formatting, such as the font used and the shading type used for each - e.g. PW. These are currently on the following page (L134-138) so not obvious to the reader.

Added the explanation of keys and figure reworked according to other changes (removing of golf site). Also added the location of the nests as black squares.

Section 2.2 - The breakdown of hedgehogs equals 73, although 72 is stated on the first line. As 12 were rehabilitated it isn't correct to say on the first line that 72(73) were captured - perhaps studied/observed/radiotracked?

Changed

Spelling/wording errors - e.g. L112 & L242 - weighted/weighting; L145 Rubish; L163 at the moment of the year - should read the time of year or season. L164 - Apr not Abr; L185 & L259 - built not build.

Thank you for the corrections. Done.

L118 & L123 - The attachment of GPS tags is described but not VHF tags

Both were the same methodology, as we removed the GPS part, we passed the description to the VHF.

L128 - how can you ensure optimal health and stress conditions? How was this measured? I would suggest this can only be determined by veterinary examination. Perhaps this should be reworded if this isn't what was done.

No, a veterinary examination was not done. But we assured the individuals had no injuries and were calm when released. Rewritten.

L207 - RSF needs to be defined for clarity.

Done

L218-223 - the use of abbreviations makes this rather confusing. I found myself needing to refer back pages to know what was being discussed.

Added the meaning of the abbreviations in text, but maintained them in brackets so that it is easier to know to what it is referring in the tables.

L222/L240 - *used significantly more

Changed

Reviewer 2 Report

Comments and Suggestions for Authors

Author Response

1. In “1. Introduction”, why was this study conducted? What is the significance of

the study?

Added some lines addressing the significance of the study.

2. In “1. Introduction”, there are a few too many paragraphs. It is recommended that

the paragraphs be merged and streamlined.

Merged some of the paragraphs

3. In line 100, why use data from 4 years ago?

Why not? This is part of a PhD thesis and the sampling was done in this period. The data collected were also used to publish other articles.

4. In “3. Results”, it is suggested that adding some graphs would allow for a more

visual presentation of the results.

We already discussed how the data should be represented and tried several things. The sheer amount of different data to represent graphically made it very confusing to interpret graphical representations. For instance, we would need 4 graphs for macrohabitats and 4 graphs for subhabitats, one for each grouping, because fitting all in one or two is too messy. Then put somehow the statistics on the graph, or we would need a table anyway. We think the way it is is the most understandable and cleaner way to represent our results.

Thank you for your comments.

Reviewer 3 Report

Comments and Suggestions for Authors

Finding the nests of European hedgehogs in the pine forests on the golf course is obvious because they are not going to use areas completely free of protection such as the green. I think that the choice of this habitat should be justified and the results reinterpreted, commenting on this characteristic.

The discovery of the two species of hedgehogs in the same nest also deserves a specific comment given the surprising and interesting nature of this behavior. Also considering that the two specimens are males.

Author Response

Finding the nests of European hedgehogs in the pine forests on the golf course is obvious because they are not going to use areas completely free of protection such as the green. I think that the choice of this habitat should be justified and the results reinterpreted, commenting on this characteristic.

Given the comment and concerns of reviewer #1 on the golf course data, we decided to remove this part entirely.

The discovery of the two species of hedgehogs in the same nest also deserves a specific comment given the surprising and interesting nature of this behavior. Also considering that the two specimens are males.

We agree, this was also an unexpected encounter for us. In fact we mention it several times along the manuscript, and now added it also to the abstract.